# Investigation of the Tribological Behavior and Microstructure of Plasma-Cladded Fe–Cr–Mo–Ni–B Coating

**DOI:** 10.3390/ma15196595

**Published:** 2022-09-23

**Authors:** Junfu Chen, Fenglong Zhang, Xianghui Ren, Yaoshen Wu, Shanguo Han, Manxia Cai, Zhenglong Li, Likun Li

**Affiliations:** 1Guangdong Provincial Key Laboratory of Advanced Welding Technology, China-Ukraine Institute of Welding, Guangdong Academy of Sciences, Guangzhou 510650, China; 2Faculty of Intelligent Manufacturing, Wuyi University, Jiangmen 529020, China

**Keywords:** Fe–Cr–Mo–Ni–B, Mo_2_FeB_2_, wear resistance, microstructure, plasma cladding

## Abstract

In this study, an Fe–Cr–Mo–Ni–B coating was prepared using plasma cladding on Cr5 steel substrate. The microstructure, phase evolution and tribological performance of the Fe–Cr–Mo–Ni–B coating were investigated. The microstructure is mainly composed of Mo_2_FeB_2_, Fe_2_B, α-Fe, γ-Fe and MoB. The process of phase evolution in the coating was observed in situ by HT-CLSM. The Mo_2_FeB_2_ phase with good thermodynamic stability can exist in the high-temperature liquid phase. It also has a phenomenon of connection and merging and turns into different morphology during the plasma cladding process. The hardness value of coating was much higher than the base metal, and the hardness value of Mo_2_FeB_2_ (785.5 HV) was higher than the eutectic matrix (693.2 HV). The wear mechanisms of the cladding under dry sliding were primarily caused by adhesive wear, accompanying slight oxidation wear. The Mo_2_FeB_2_ phase has an important effect on the wear resistance property.

## 1. Introduction

The development of wear-resistant materials is driven by complex and adverse working conditions. Various cladding techniques have been used to prepare wear-resistant coatings on the surface of workpieces to repair and strengthen the workpieces. Plasma cladding is one of the most promising techniques for surface modification [1,2]. Due to the wear-resisting performance of cladding depending on the type, size and distribution of reinforcement phases, the cladding can be prepared on the surface of the base material according to the performance requirements. At present, carbides and borides are commonly used as reinforcement phases of iron-based wear-resisting coatings [3,4,5]. Boride has been widely applied to the iron and steel industry because of its higher hardness, thermal stability and excellent wear resistance. The hardness and wear resistance of the alloy can be greatly improved by using boride’s hard phase as the main reinforcing particles. Zhuang et al. [6] reported that the volume fractions and morphologies of boride in the Fe–B–C surfacing alloys could be regulated by adjusting the B and C contents. The wear resistance of the coatings with fishbones such as Fe_2_B and daisy-like Fe_3_ (C, B) is inferior to the primary rod-like Fe_2_B, which is easy to crack. Fe–Cr–B–C alloy was studied widely because the fracture toughness is improved owing to the entry of Cr into Fe_2_B [7]. The wear resistance increases first and then decreases as the Cr content increases. When the wear resistance of alloy was the best, the chromium content was 12 wt.%, and the hardness of the alloy reached the highest value of 61.1 HRC [8]. Binary boride easily reacts metallurgically with the other alloying elements, which is the common method for the synthesis of ternary borides. Mo–Fe–B and Mo–Ni–B are ternary boride systems that have been widely studied in recent years [9,10]. Zhang et al. [11] reported that ternary boride-based cermets exhibited excellent mechanical properties with fine-grained Mo_2_FeB_2_ cermets, which were prepared by the two-step sintering technique. Mo_2_FeB_2_ showed outstanding wear resistances over wide temperature ranges under dry sliding [12]. Yu et al. [13,14] investigated the effect of C content and the Mo/B atom on the performance of Mo_2_FeB_2_ metal-based ceramics. The hardness and toughness of Mo_2_FeB_2_ metal-based ceramics were the highest when the C content was 0.5%, and the transverse rupture strength shows a maximum value of 1.9 GPa at a Mo/B atomic ratio of 0.9. The hardness decreased with the Mo/B atom ratio from 0.8 to 1.1, and the strength reached the highest when the Mo/B atom ratio was 0.9. The suitable amount of boron addition into the raw material was around the range of 1–2 wt.% [15]. At present, most Mo_2_FeB_2_ cermets are prepared by reaction sintering, and few studies of Mo_2_FeB_2_-reinforced iron base coatings were reported. Xu et al. [16] reported that the wear resistance of Mo_2_FeB_2_ coating prepared by argon-arc cladding is better than Mo_2_FeB_2_ cermets prepared by vacuum sintering. Mo_2_FeB_2_ coating prepared by an arc welding method has good diffusion and metallurgical bonding with the base material, and there are no obvious pores. Although the microstructure with different component content was already investigated, the distribution and morphology of Mo_2_FeB_2_ particles prepared by cladding technology were complicated and variable, the distribution and form of particles especially are difficult to control, and good forming and performance cannot be guaranteed. Therefore, it is necessary to reveal the phase evolution and the growth mechanism of Mo_2_FeB_2_ particles. Moreover, the nucleation and growth of boride is related to quantity and distribution of the powders; this work invented a novel Fe–Cr–Mo–Ni–B powder prepared by vacuum gas atomization which reduces the inhomogeneity of the distribution.

In the present work, a novel Fe–Cr–Mo–Ni–B wear-resistant coating was fabricated on Cr5 steel by plasma cladding technology. The formation mechanism of the Mo_2_FeB_2_ particles and binary boride hard phase within the coating was elucidated. The hardness and wear mechanism of boride-reinforced Fe-based cladding were also investigated.

## 2. Materials and Methods

Annealed Cr5 steel with a size of 200 mm × 100 mm × 15 mm was used as a substrate. The Fe–Cr–Mo–Ni–B alloy powders prepared by actual air atomization method were used as cladding powders. The morphology of cladding powders is shown in Figure 1. The chemical composition of cladding alloy powder and Cr5 steel substrate are given in Table 1. The alloy powder was dried at 120 °C for 2 h in the vacuum drying oven, and the cladding surface of the Cr5 steel substrate was ground with an angle grinder to remove the rust layer and cleaned with alcohol and acetone to avoid deposition before the test. A plasma cladding machine (Castolin Eutectic, Kriftel, Germany) with a provision for synchronized feeding was used. Pure argon was served as the powder feed gas and shielding gas. The following conditions were maintained during the plasma cladding process: a current of 100 A, powder feeding rate of 8.5 g/min and scanning speed of 3 mm/s. The schematic diagram of the plasma cladding operation was shown in Figure 2.

The clad specimens were cut into cross-section and longitudinal section with a size of 10 mm × 20 mm by using wire electrical discharge machining, using a vernier caliper to measure the size. Then, these specimens were pre-treated using the standard metallographic techniques and etched using the corrosive solution (HF, HNO_3_ and H_2_O with a ratio of 1:1:25) at room temperature for 90 s. The microstructure of the coating was observed by OMLPUS BX-6 metallographic microscope and the field emission scanning electron microscope (FEI, QUANT250, Eindhoven, NLD and ZEISS, Gemini SEM300, Oberkochen, GER) equipped with an energy dispersion spectroscope (Noran System 7, ThermoFisher Scientific, MA, USA). A high-resolution X-ray diffraction (SmartLab 9 KW, Rigaku, Tokyo, JPN) with a step size of 0.02° and 2θ range from 10° to 90° was used to identify the phase composition of the coating. 

Evolution of phase composition in the coating was observed in situ by high-temperature confocal scanning laser microscopy (HT-CLSM). The Fe–Cr–Mo–Ni–B coating was machined into a cylindrical specimen with the size of Φ 5 mm × 3 mm. Before heating, the HT-CLSM chamber was filled with argon in order to avoid the oxidation phenomenon, and the specimen was prepared by mechanical polishing. The sample was heated following the heating cycles with a rapid heating rate of 500 °C/min from 100 °C to 1150 °C because the phase transition temperature ranged from 1012.9 °C to 1149 °C, according to the DSC curve of the Fe–Cr–Mo–Ni–B coating, as shown in Figure 3.

The Rockwell hardness was tested by the Rockwell hardness tester (HR-150A, Shandong Laizhou Hardness Tester factory, Shandong, China) with a load of 150 kg. Different areas on the surface of the Fe–Cr–Mo–Ni–B coating were selected for 8 tests, and the average value after removing the maximum value and the minimum value were taken. The microhardness was measured by Vickers microhardness tester (Wilson VH1202, Fort Worth, TX, USA) with a load of 100 g and a loading duration of 15 s. A ball-on-disk wear tester (UMT-3, CETR, Campbell, CA, USA) was used to evaluate the wear resistance of the cladding under dry condition at room temperature. The loads applied on the counterpart were 20, 30, 40 and 50 N. GCr15 balls with a diameter of 5 mm were selected as counterparts. The frequency of reciprocating wear, sliding speed and wear time were 6 Hz, 60 mm/s and 30 min, respectively, because after 30 min there was little change according to the results of the pre-experiment. The worn surfaces of clad samples were examined by Bruker direct reading spectrometer (Q4 TASMAN, BRUKER, Berlin, Germany) to obtain wear size and calculate wear rate. Morphology of the worn surfaces was observed by scanning electron microscope.

## 3. Results and Discussion

### 3.1. Microstructure

The SEM image of the Fe–Cr–Mo–Ni–B coating and EDS sweeping analysis area are shown in Figure 4, and the element contents of various regions are given in Table 2. The main elements of the blocky phase in point 1 are 55.78 at.% Mo and 27.61 at.% Fe, and the atomic ratio of Mo and Fe is about 2:1. It can be inferred that the blocky phase in the Fe–Cr–Mo–Ni–B coating is the precipitated Mo_2_FeB_2_ ternary boride. In point 2 and 3, the eutectic matrix structure mainly contains the Fe element and a small amount of the Cr, Mo and Ni elements. The XRD pattern of the Fe–Cr–Mo–Ni–B coating is shown in Figure 5. It is clearly indicated that the phase of the coating mainly comprises the α-Fe, γ-Fe, Mo_2_FeB_2_, Fe_2_B and MoB. The observed microstructure of the Fe–Cr–Mo–Ni–B coatings are shown in Figure 6. It can be seen that the cladding is metallurgically bonded to the substrate, and the bonding interface is smooth without cracks and pores. The microstructure of the Fe–Cr–Mo–Ni–B coating is mainly composed of the Mo_2_FeB_2_ phase distributed in the eutectic matrix. The grain size of the Mo_2_FeB_2_ phase is mostly between 5–30 μm. Due to the rapid heating and cooling rate of the plasma cladding process, the agglomerated, elongated and square Mo_2_FeB_2_ phase in the eutectic matrix is different from that in the cermets [17]. According to the previous studies [18,19], with the increase in B and Mo content, the volume fraction of Mo_2_FeB_2_ increased significantly and irregular coarse Mo_2_FeB_2_ appeared. In this study, Fe–Cr–Mo–Ni–B powders were prepared by vacuum gas atomization. So, the excessive cooling rate leads to insufficient element diffusion and uneven distribution of component concentrations in the molten pool. The square Mo_2_FeB_2_ phase is formed because of the same growth rate in growing directions. The Cr element could significantly decrease the hard phase grain size, and the Mo_2_FeB_2_ phase may inhibit the growth of the hard phase particles in the length direction [20]. The particle morphology was gradually transformed from elongated to nearly equiaxed, while the lattice constant of the Mo_2_FeB_2_ phase changed. However, with the high cooling rate of the molten pool, the nucleation rate of the Mo_2_FeB_2_ phases increases. A large number of elongated and irregular coarse Mo_2_FeB_2_ particles are observed in the coating. During the rapid cooling, the Mo_2_FeB_2_ particles would grow to connect to each other and turn into irregular, coarse hard phase due to an uneven distribution of temperature in the molten pool. The same crystal structure strengthens the connection between the Fe_2_B and Mo_2_FeB_2_ phases. With the decrease in the Mo_2_FeB_2_ phase, the volume fraction of the eutectic structure which formed by Fe and B elements increases substantially. The volume fraction of the Mo_2_FeB_2_ phase and the eutectic boride is negatively correlated, and the microstructure is complicated and varied.

The elemental distribution maps of the Fe–Cr–Mo–Ni–B coating are shown in Figure 7. Mo, Fe, Ni and Cr are all contained in the Mo_2_FeB_2_ phases and eutectic matrix. Due to the low solid solubility of B in austenite and ferrite, B is mainly used for the formation of boride phase rather than dissolution in the matrix. The addition of alloying elements will improve the solid solubility of B [21]. Therefore, B is rarely dissolved in the iron matrix. With the increase in Mo, the volume fraction of Mo_2_FeB_2_ increased significantly and an irregular Mo_2_FeB_2_ phase appeared. B and Mo are mainly consumed during the formation of the Mo_2_FeB_2_ phase. Fe is mainly distributed in the eutectic structure and matrix, while a small amount of Cr, Mo and Ni are distributed in the iron matrix. The Cr and Ni elements are evenly distributed in the Fe–Cr–Mo–Ni–B coating. Cr and Ni are considered to partially substitute Mo and Fe in the Mo_2_FeB_2_ phase to form complex borides [22]. The ternary boride M_3_B_2_ formed by the dissolution of Cr and other alloying elements into the Mo_2_FeB_2_ phase. Ni is often dissolved in the iron matrix as an austenite-forming element, which is conducive to ensuring the strength of the coating and improving the plasticity and toughness of the Fe–Cr–Mo–Ni–B coating. Due to the replacement and solid solution of Mo, Cr, Ni and other alloyed elements, the lattice distortion of the borides Mo_2_FeB_2_ and Fe_2_B is induced to form multiple composite borides M_3_B_2_ and M_2_B, and the mechanical properties are improved.

The thermal analysis results of the Fe–Cr–Mo–Ni–B coating are shown in Figure 8. According to Figure 8, the melting of the phase before 800 °C is endothermic, followed by exothermic decomposition. It also can be seen that the phase transition occurs at 1012.9 °C. In situ HT-CLSM micrographs illustrating the transformation of the microstructure in the Fe–Cr–Mo–Ni–B coating are presented in Figure 9. With the increase in temperature, the reaction γ+Fe_2_B→L occurs at about 1079 °C, as shown in Figure 9a. The Fe_2_B phases connect to each other and turn into an eutectic network matrix, and the network breaks during heating (Figure 9b–f). It can be seen that a certain amount of tiny Mo_2_FeB_2_ particles are coexisting with the liquid phase at high temperature. The growth and coalescence process of the eutectic matrix and the Mo_2_FeB_2_ grains could occur at about 1150 °C. The borides were precipitated from the molten pool, and the growth of the Mo_2_FeB_2_ crystal is controlled by the mechanism of Ostwald ripening during the plasma cladding. This phase of coating, namely γ-Fe, Mo_2_FeB_2_, Fe_2_B and MoB, plays an important role in the aggregation and rearrangement of the Mo_2_FeB_2_ grains [23].

The Fe–Cr–Mo–Ni–B powder designed in this report is probably a hypoeutectic component alloy according to the Fe–B phase diagram. The alloy powders melt to form a melting pool with increasing temperature using a plasma arc heat source. According to thermodynamic theory, the reaction can proceed spontaneously only if the free energies ΔG_T_ < 0 are satisfied. Jun [24] calculated that the formation of the free energies of MoB and Fe_2_B were all less than zero, and the absolute value was large in the Mo–Fe–B ternary alloy system, which indicated that they can be formed in the weld pool, and Mo_2_FeB_2_ can exist stably in high temperature. Therefore, the main chemical reactions that exist in the high-temperature weld pool during the plasma cladding process are as follows:



(1)
Mo+B=MoB


(2)
$${2\mathrm{Fe}+B=\mathrm{Fe}}_{2}B$$


(3)
$${\mathrm{Fe}+2\mathrm{MoB}=\mathrm{Mo}}_{2}{\mathrm{FeB}}_{2}$$


(4)
$${2\mathrm{Mo}+2\mathrm{Fe}}_{2}{B=\mathrm{Mo}}_{2}{\mathrm{FeB}}_{2}+3\mathrm{Fe}$$



According to the binary phase diagram of Fe–B and Mo–B, the precipitation temperature of molybdenum and boron compounds is much higher than that of Fe–boron compounds, which could prove that the Mo_2_FeB_2_ hard phase is preferred to form and stably exist at the high-temperature liquid phase [25]. Hence, the Mo_2_FeB_2_ crystal nucleus begins to precipitate from the liquid phase by consuming a large amount of Mo and B. The results of HT-CLSM proved that the coexistence of the liquid phase and Mo_2_FeB_2_ reached equilibrium. Cr and Ni atoms with a high concentration in the liquid phase are replaced by Mo_2_FeB_2_ to precipitate the positions of Mo and Fe in the crystal to form the compound ternary boride M_3_B_2_. Due to the low solid solubility of the B element in austenite and ferrite, the B atom is repellent and diffuses to the liquid phase, so that the concentration of B in the liquid phase increases to the eutectic point, and the eutectic reaction L→γ+Fe_2_B occurs. As the temperature decreases continuously, the γ-Fe phase is almost completely precipitated around boride grains, and the transformation of γ→α+ Fe_2_B occurs. The resulting boride Fe_2_B interacts with other alloying elements such as Cr, Mo and Ni in the molten pool and finally forms the boride hard phase such as M_2_B. Due to the fast cooling rate, the unreacted alloying elements are dissolved into the matrix to form Fe (Cr, Mo and Ni). This solid solution of alloying elements improves austenite stability, so that a part of γ-Fe, rich in Cr, Mo and Ni, is retained without undergoing an allotropic transformation from γ-Fe to α-Fe [26]. It is beneficial to enhance the toughness of the Fe–Cr–Mo–Ni–B coating. 

### 3.2. Vickers Hardness and Tribological Properties

Figure 10 shows the curves of Vickers hardness along the longitudinal section and cross-section of the Fe–Cr–Mo–Ni–B coating, respectively. The Fe–Cr–Mo–Ni–B coating, the heat-affected zone and the substrate can be distinguished by the microhardness variation. The hardness of the Fe–Cr–Mo–Ni–B coating is much higher than that of the base metal. The uneven size and distribution of the Mo_2_FeB_2_ hard phase of the Fe–Cr–Mo–Ni–B coating results in the fluctuation of the microhardness. The hardness is 785.5 HV, where the Mo_2_FeB_2_ hard phase is agglomerated, and the hardness of the eutectic matrix is 600.6 HV. The average Vickers hardness of the Fe–Cr–Mo–Ni–B coating is 693.2 HV. Therefore, the hardness improvement of the Fe–Cr–Mo–Ni–B coating mainly comes from the formation of the Mo_2_FeB_2_ phase. The boride hard phase can improve the hardness and wear resistance of the Fe–Cr–Mo–Ni–B coating, and the matrix can guarantee the toughness of the Fe–Cr–Mo–Ni–B coating, which effectively prevents the spalling of the Mo_2_FeB_2_ phase. 

Wear scar size, wear volumes and wear rate of the Fe–Cr–Mo–Ni–B coating under the four different loads are listed in Table 3. The wear rate can be calculated by the following Equation [27]:(5)$$W_{r}=\frac{V}{F\times D}$$

Where V is the wear volume, *F* is the load in wear test and *D* represents the Sliding distance in the wear test. The friction coefficient and the microstructure of the worn surfaces of the cladding are shown in Figure 11 and Figure 12. It can be seen that the wear rate of the Fe–Cr–Mo–Ni–B coating increases gradually with the increase in load. Due to the existence of a large number of protruding particles such as the Mo_2_FeB_2_ phase on the surface of the Fe–Cr–Mo–Ni–B coating, the grinding ball impinged with the Mo_2_FeB_2_ particles at the initial stage of friction and wear. The curve of friction coefficient tends to flatten after a rapid increase. With the increase in the load, the average friction coefficient of the Fe–Cr–Mo–Ni–B coating decreases gradually due to the formation of an oxide layer and increase in the contact area between the grinding ball and the Fe–Cr–Mo–Ni–B coating. To better describe the wear mechanism, a schematic illustration of the wear process under different loads is shown in Figure 13. When the loads are 20 N and 30 N, there is no obvious spalling on the surface of the Fe–Cr–Mo–Ni–B coating, and the friction coefficient is relatively gentle. The oxide layer can be formed by thermal oxidation and is difficult to lubricate the worn surface with. When the loads increase to 40 N and 50 N, severe plastic deformation occurred in the Fe–Cr–Mo–Ni–B coating. Adhesive shear action occurs between the iron matrix and the grinding ball, which will lead to the increase in the friction coefficient. The debonding of the boride phase and the softening of the binder phase may increase the friction force of the friction pairs. Therefore, the curve of the friction coefficient appears as a rising wave peak in the early wear stage, and then an oxide layer may appear, decreasing the friction coefficient significantly. The iron matrix is preferentially peeled off as abrasive debris, most of which are pushed out of the wear scar by the friction pair, and a small part is left as abrasive particles. The spalling of the eutectic matrix results in the Mo_2_FeB_2_ phase protruding. The debris and the protruding Mo_2_FeB_2_ particles are relatively wear-resistant, resulting in great fluctuations in the friction coefficient.

Under 20 N and 30 N, the wear surface of the Fe–Cr–Mo–Ni–B coating shows shallow wear marks along the sliding direction and slight adhesive wear. Due to the existence of large areas of boride hard phases in the matrix, such as M_3_B_2_ and M_2_B, micro-cutting of the Fe–Cr–Mo–Ni–B coating is hindered. The hardness of GCr15 is higher than that of the coating, so the grinding ball would easily penetrate into the surface and produce wear volume loss. Plastic deformation and adhesion lead to the spalling of the eutectic matrix. With the increase in temperature in the friction process, slight oxidation was found on the worn surface. Under 40 N and 50 N, the spalling of eutectic matrix is more serious and some of the brittle M_3_B_2_ phase breaks under extrusion. The connection between the eutectic borides and M_3_B_2_ enhance the bonding force of the M_3_B_2_ phase. The M_3_B_2_ phase plays an important role in resistance to friction and wear. Therefore, there is plastic deformation and the formation of friction oxides on the worn surface. Additionally, the wear mechanism includes adhesive wear, slight oxidation wear and abrasive wear.

## 4. Conclusions

In this study, the formation mechanism, morphology, distribution position and size of various forms of boronized phases such as Mo_2_FeB_2_ boride particles and binary borides in Fe–Cr–Mo–Ni–B coating are studied, as well as the effect of borides on the hardness and wear resistance of the Fe–Cr–Mo–Ni–B coating. The main research conclusions are as follows:(1)The Fe–Cr–Mo–Ni–B coating was prepared by plasma cladding technology. The Mo_2_FeB_2_ hard phase connects with other phases and grows to a different morphology.(2)The microstructure of the Fe–Cr–Mo–Ni–B coating remains stable when the temperature is below 1050 °C. The Mo_2_FeB_2_ phase has good thermodynamic stability and stably exists in the high-temperature liquid phase. The growth and coalescence process of the eutectic matrix and Mo_2_FeB_2_ grains could occur at about 1150 °C.(3)The hardness value of the Fe–Cr–Mo–Ni–B coating was much higher than that of the base metal. The wear mechanisms of the Fe–Cr–Mo–Ni–B coating under dry sliding were primarily caused by adhesive wear, accompanying abrasive wear and oxidation wear, and the Mo_2_FeB_2_ phase plays an important role in the wear-resistance property.

## Figures and Tables

**Figure 1 materials-15-06595-f001:**
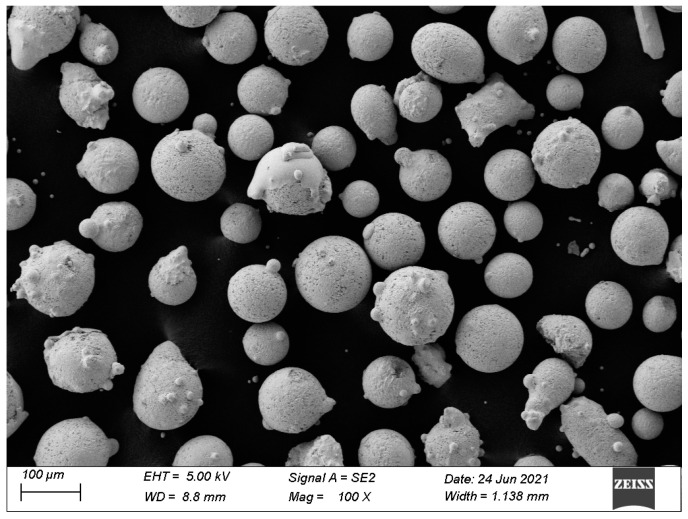
SEM image of Fe–Cr–Mo–Ni–B powder.

**Figure 2 materials-15-06595-f002:**
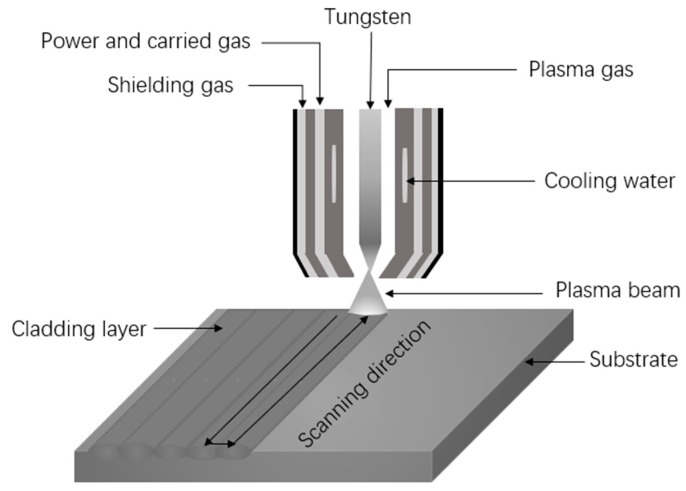
Schematic diagram of plasma cladding.

**Figure 3 materials-15-06595-f003:**
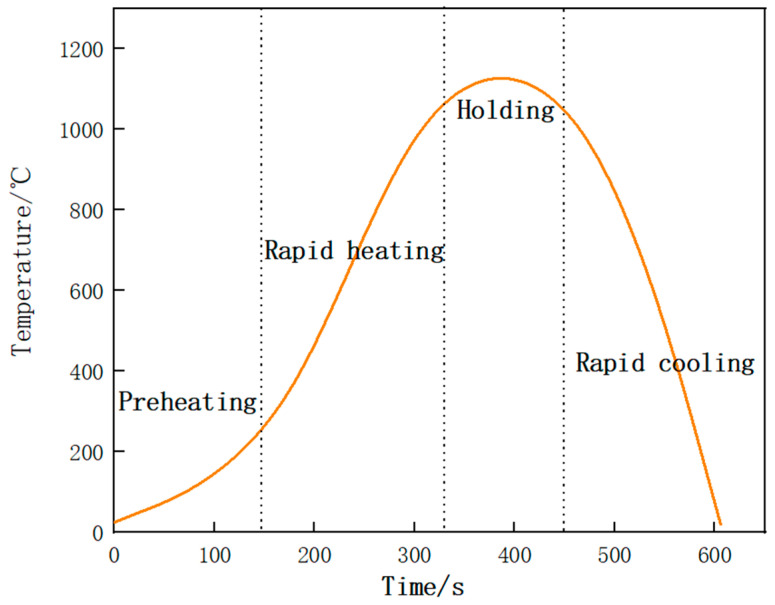
Temperature versus time curve.

**Figure 4 materials-15-06595-f004:**
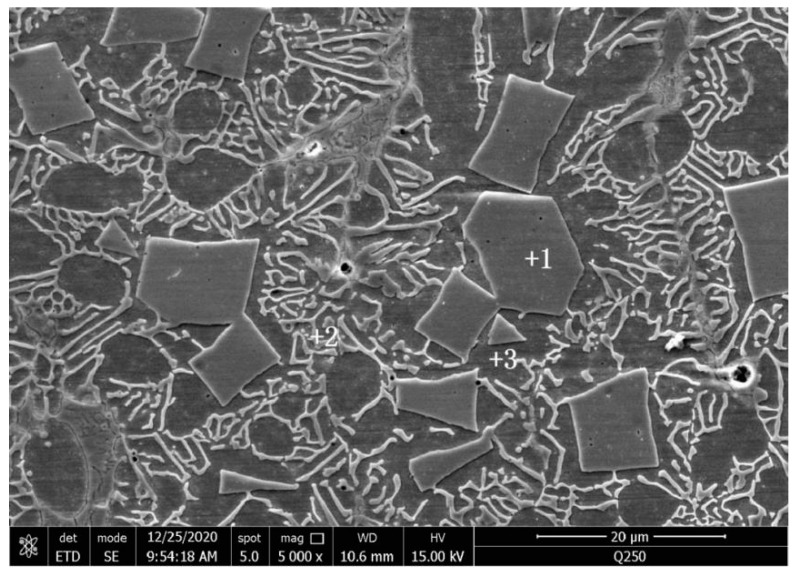
SEM image of the Fe–Cr–Mo–Ni–B coating.

**Figure 5 materials-15-06595-f005:**
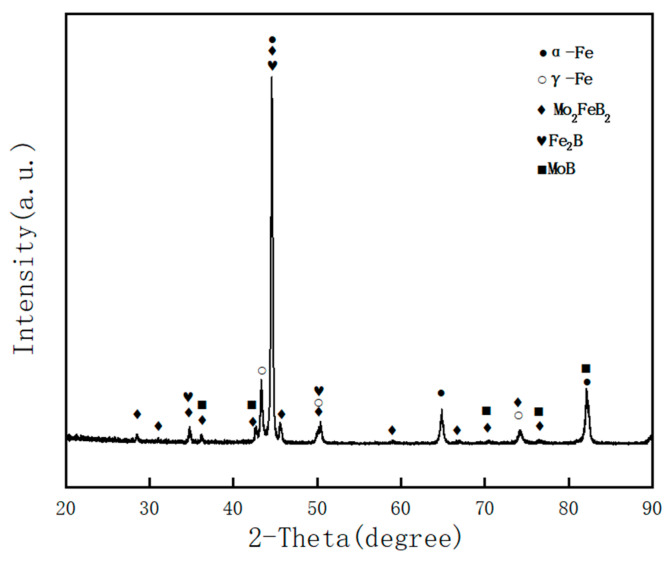
XRD pattern of the Fe–Cr–Mo–Ni–B coating.

**Figure 6 materials-15-06595-f006:**
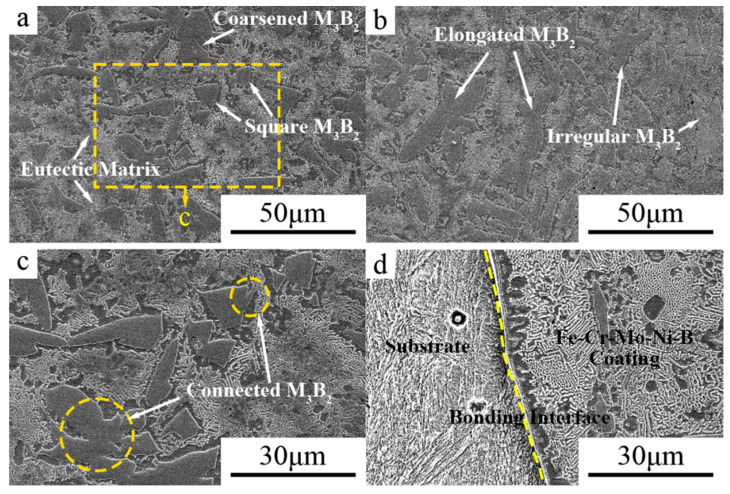
OM image of the Fe–Cr–Mo–Ni–B coating: (**a**) block morphology; (**b**) strip morphology; (**c**) high magnification of block morphology; (**d**) fusion area.

**Figure 7 materials-15-06595-f007:**
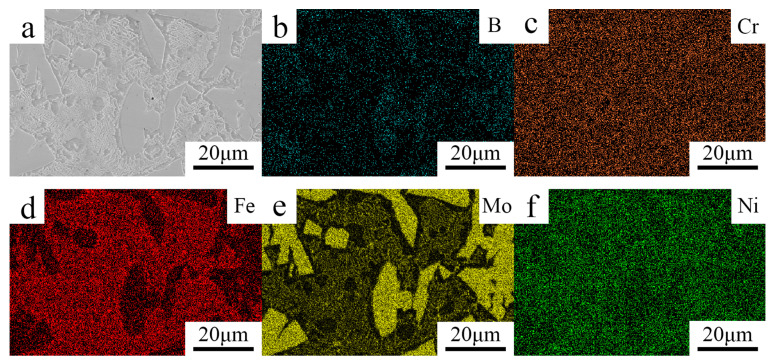
Elemental distribution maps of the Fe–Cr–Mo–Ni–B coating: (**a**) SEM image; (**b**) B; (**c**) Cr; (**d**) Fe; (**e**) Mo; (**f**) Ni.

**Figure 8 materials-15-06595-f008:**
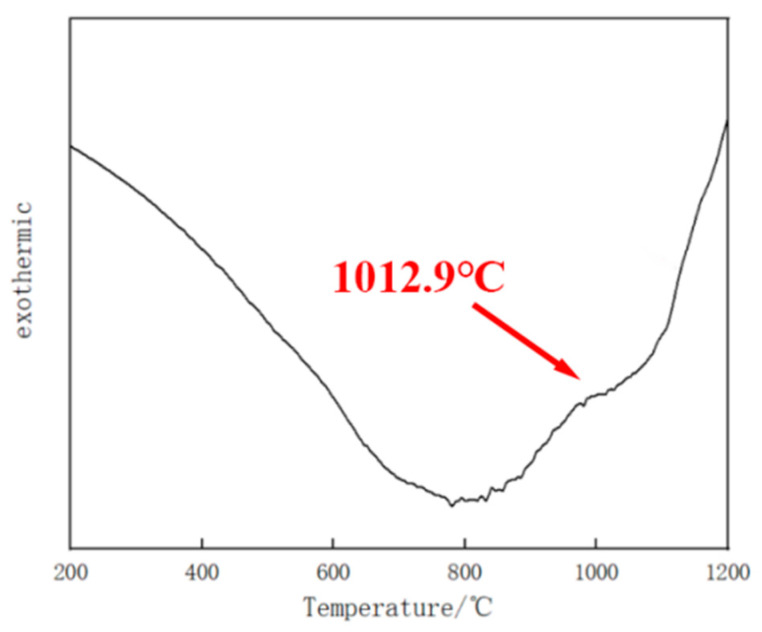
DSC curve of the Fe–Cr–Mo–Ni–B coating during the heating process.

**Figure 9 materials-15-06595-f009:**
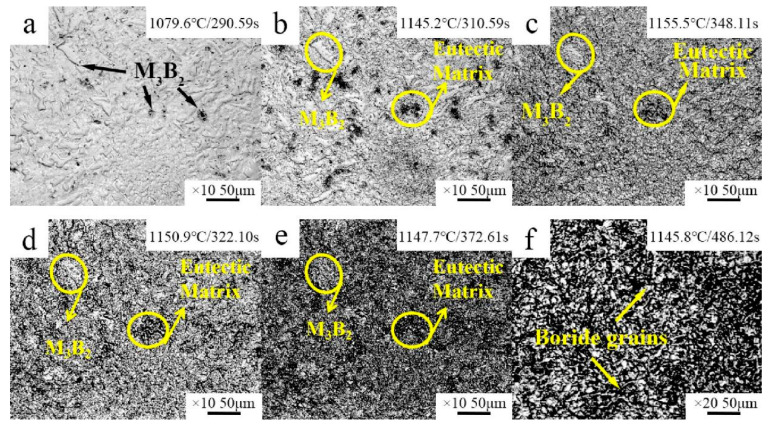
HT-CLSM snapshots of partial remelting of the Fe–Cr–Mo–Ni–B coating: (**a**) 1079.6 °C/290.59 s; (**b**) 1145.2 °C/301.59 s; (**c**) 1155.5 °C/348.11 s; (**d**) 1150.9 °C/322.10 s; (**e**) 1147.7 °C/372.61 s; (**f**) 1145.8 °C/486.12 s.

**Figure 10 materials-15-06595-f010:**
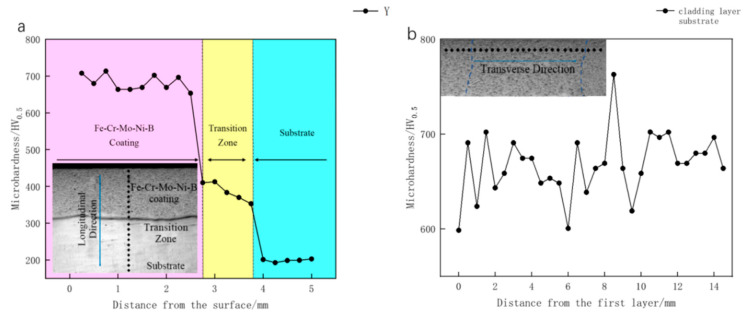
Microhardness of the Fe–Cr–Mo–Ni–B coating: (**a**) longitudinal section; (**b**) cross-section.

**Figure 11 materials-15-06595-f011:**
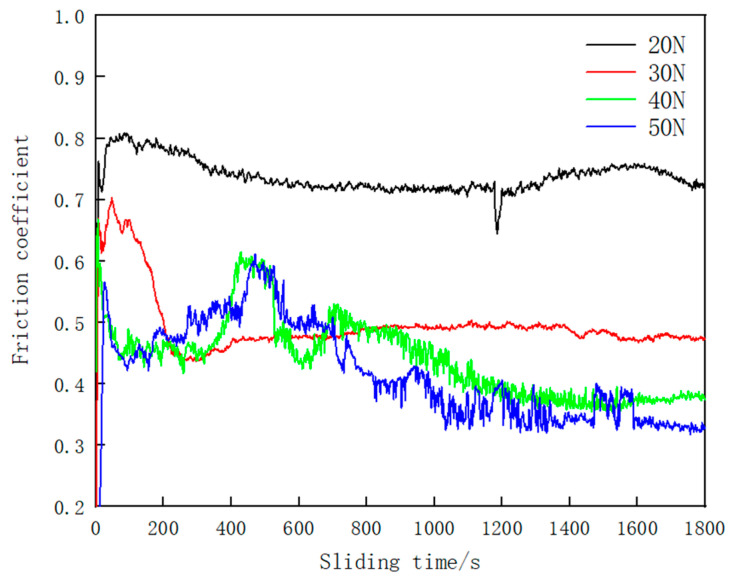
Friction coefficient of the Fe–Cr–Mo–Ni–B coating.

**Figure 12 materials-15-06595-f012:**
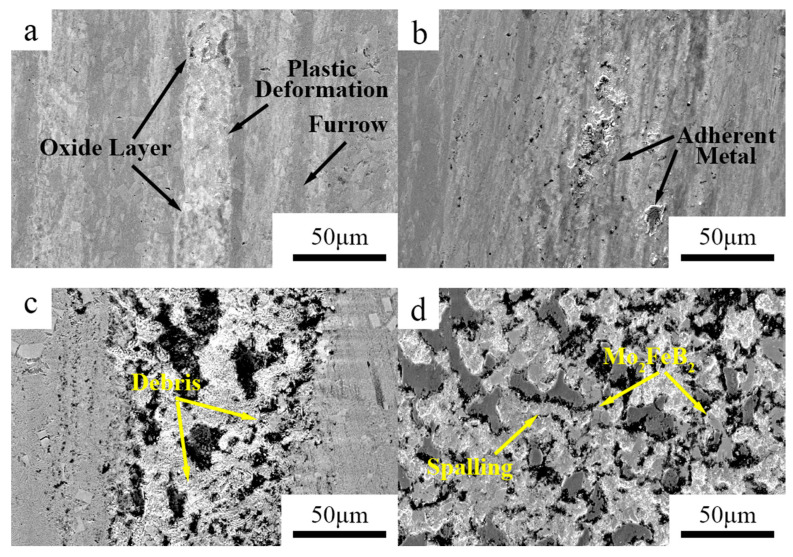
SEM images of worn surfaces on the Fe–Cr–Mo–Ni–B coating under different load: (**a**) 20 N; (**b**) 30 N; (**c**) 40 N; (**d**) 50 N.

**Figure 13 materials-15-06595-f013:**
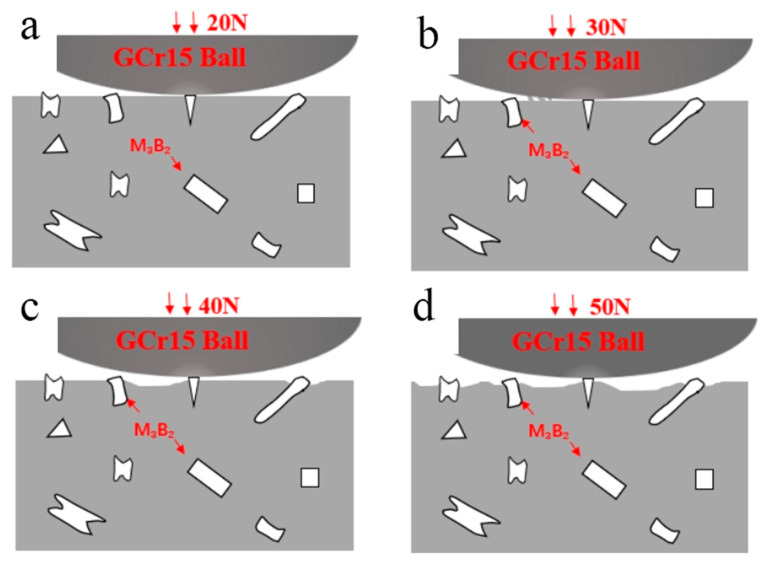
Schematic diagram of the wear process under different loads on the Fe–Cr–Mo–Ni–B coating: (**a**) 20 N; (**b**) 30 N; (**c**) 40 N; (**d**) 50 N.

**Table 1 materials-15-06595-t001:** Chemical compositions of the alloy powders/Cr5 steel (wt.%).

Element	Cr	Mo	Ni	Si	Mn	P	S	B	Fe
Base metal	5	0.5	0.20	0.12	0.42	≤0.03	≤0.03	-	balance
Alloy powders	4	30	6	-	-	-	-	2	balance

**Table 2 materials-15-06595-t002:** Element contents of various regions.

Element	Point 1		Point 2		Point 3	
	wt.%	at.%	wt.%	at.%	wt.%	at.%
Fe	19.86	27.61	71.83	75.15	78.58	80.20
Cr	10.47	15.63	11.00	12.36	9.69	10.62
Mo	68.93	55.78	11.92	7.26	5.85	3.48
Ni	0.74	0.98	5.25	5.23	5.88	5.71

**Table 3 materials-15-06595-t003:** Wear scar size and wear rate.

Load	Width of Wear Scar/mm	Wear Volume /mm^3^	Wear Rate/(mm^3^·N^−1^·m^−1^)	Friction Coefficient
20 N	0.69	0.00245	1.134 × 10^−6^	0.7375
30 N	0.84	0.00568	1.753 × 10^−6^	0.4950
40 N	0.92	0.00771	1.784 × 10^−6^	0.4408
50 N	1.08	0.01172	2.171 × 10^−6^	0.4178

## Data Availability

The data that support the findings of this study are available from the corresponding author upon reasonable request.

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
