# Peer review of "Investigation of the Tribological Behavior and Microstructure of Plasma-Cladded Fe–Cr–Mo–Ni–B Coating"

_materials, 2022, doi:10.3390/ma15196595_

Round 1
Reviewer 1 Report
English Should be improved
Please, read my notations in the manuscript. The friction surface examinations should be continued. Based on new studies, reexplain the wear mechanism. Special attention should be paid to the coefficient of friction because typically increase in load leads to an increase in CoF, and in Your case, it is vice-versa. Also, rearrange the manuscript as I recommend.
I think, that after the manuscript should be rejected and reconsidered after major revision and additional topographical studies

Reviewer 2 Report
Dear Authors, I think that the topics covered fit well into the current research scenario and the novelty of your work is well highlighted in the introduction. However, the article should be revised to clarify some neglected methodological aspect, which also affect the presentaion of the results: Line 71 please, give an indication of the cladding thickness and how it was measured. Line 79 please, specify if you mean a cylindrical specimen. Line 85 you stated the use of a Rockwell hardness tester, but the Rockwell hardness. values are missing from the results presentation. In any case this type of measurement is not suitable for thin thicknesses.Table 2 for a useful comparison, wear rate of the base material should be given as well. In this way, the effects of cladding on wear behavior could be better highlighted.
Fig. 10 the method for calculationg friction coefficent has not previously specified.
Line 204 the method for calculating wear volumes has not been described.
As for the English language, it needs to be refined. Some hints are listed below: Line 24 correct “depending” in “depends” Line 25 correct “use” in “used” Line 31 correct “wildly” in “widely” Line 33 correct “with” in “as” Line 37 correct “prepared” in “were prepared” Line 49 correct “relate” in “is related” Line 50 correct “powders” in “powder” Line 50 correct “reduce” in “reduces” Line 57 correct “dimension” in “size” Line 57 correct “to the substrate” in “for the substrate” Line 58 correct “served” in “used” Line 71 correct “cladding specimens” in “clad specimens” Line 79 correct “dimensions” in “size” Line 88 correct “in room temperature” in “at room temperature” Line 91 correct “cladding sample” in “clad samples” Line 96 correct “was” in “is” Line 98 correct “cladding is bonded with the substrate metallurgically” in“cladding is metallurgically bonded to the substrate”
Line 100 correct “mostly” in “is mostly” Line 102 correct “with that” in “from that” Line 108 correct “maybe inhibit” in “may inhibit”Line 139 correct “are characterized” in “are obtained”
Line 168 in this description it is preferable to use the present tense
Line 182 correct “Fig. 9 are” in “Fig. 9 shows”
Line 184 correct “than the base metal” in “than that of the base metal”
Line 185 correct “result” in “results”
Line 190 correct “prevent” in “prevents”
Line 218 correct “Severe” in “severe”
Line 227 correct “shows that there are shallow wear marks” in “shows shallow wear marks”
Line 244 correct “remain” in “remains”
Line 245 correct “exist” in “exists”
Line 248 correct “than the base metal” in “than that of the base metal”
Reviewer 3 Report
The manuscript titled “ Investigation on the Tribological Behavior and Microstructure of Plasma-Cladded Fe-Cr-Mo-Ni-B Coating” focuses on fabrication of wear-resistant coatings. The work is written understandably and topic is actual. However, the level of English should be significantly improved, ie:
1. The process of phase evolution in the coating was observed in situ by HT-CLSM. Mo2FeB2 phase with the good thermodynamic stability can stably exist in the high temperature liquid phase and grow to connect to each other and turn into different morphology during plasma cladding process.
2. The wear mechanisms of the cladding under dry sliding were primarily caused by adhesive wear, accompanying abrasive wear and oxidation wear. and the Mo2FeB2 phase has an important effect on the wear resistance property.
3. Annealed Cr5 steel with a dimension of 200 mm × 100 mm × 15 mm was used as to the substrate.
4. Then these specimens were pre-treated using the standard metallographic techniques and etched using the corrosive solution (HF, HNO3 and H2O with a ratio of 1:1:25) at room temperature for 90s.
5. Cr element could significantly decrease the hard phase grain size and Mo2FeB2 phase maybe inhibit a growth of the hard phase particles in the length direction
6. The same crystal structure strengthens the connection strength between Fe2B and Mo2FeB2 phase.
7. A range of in situ HT-CLSM micrographs are characterized by HT-CLSM illustrating the transformation of microstructure in Fe-Cr-Mo-Ni-B coating are is presented in Figure 8. The Fe2B phase connect to each other and turn into eutectic network matrix and the network breaking during heating (Fig. 8c to e). It can be seen that, certain amount of tiny Mo2FeB2 particles are coexisting with liquid phase at high temperature liquid phase.
8. Jun[23]calculated the formation free energies of MoB and Fe2B were all less than zero and the absolute value was large, which indicated that they can be formed in the weld pool and Mo2FeB2 can exist stably in high-temperature weld pool.
9. Fig.11 SEM images of worn surfaces on Fe-Cr-Mo-Ni-B coating under different load: (a) load of 20 N on, (b) load of 30 N on, (c) load of 40 N, (d) load of 50 N
10. Fig.12 Schematic diagram of the wear process under different load on Fe-Cr-Mo-Ni-B coating: (a) load of 20 N, (b) load of 30 N, (c) load of 40 N, (d) load of 50 N on
11. The Mo2FeB2 hard phase connect with each other phases and grow to different morphology.
There are also numerous editorial errors in the text (lack of space, upper letters in words, double dot, lack of comma– line 9, 10, 72, 124, 132, 194, 218, 238).
Few sentences are not clear or should be rewritten:
1. The hardness value of coating was much higher than the base metal and the hardness value of Mo2FeB2 (785.5HV) was much higher than the eutectic matrix (693.2HV). (Authors use twice “was much higher than”)
2. Xu et al.[16] reported that the wear resistance of Mo2FeB2 coating is better than Mo2FeB2 cermets prepared by the vacuum sintering. (propably author forget add welding metallurgy)
3. Cr and Ni atoms with high concentration in the liquid phase are replaced by Mo2FeB2 to precipitate the positions of Mo and Fe in the crystal to form the compound ternary boride M3B2 and the solution of alloying elements improves the mechanical property of the borides. (I don’t understand this sentence)
Additionally :
1. Figure 5 Microstructure of Fe-Cr-Mo-Ni-B coating: (a) Block morphology; (b) Strip morphology; (c) High magnification; (d) Fusion missing method (there is lack of method used)
2. The shortcut on Fig.5. HAZ should be explained below figure or omitted.
